# Improvement of Laccase Activity in Co-Culture of *Panus lecomtei* and *Sporidiobolus pararoseus* and Its Application as an Enzymatic Additive in Biomass Hydrolysis and Dye Decolorization

Rubén Darío Romero Peláez [1,2], Luana Assis Serra [2], Daiana Wischral [2], Joice Raísa Barbosa Cunha [2], Thais Demarchi Mendes [2], Thályta Fraga Pacheco [2], Felix Gonçalves de Siqueira [2,*] and João Ricardo Moreira de Almeida [1,3,*]

1   Graduate Program of Microbial Biology, Department of Cell Biology, Institute of Biology, University of Brasilia, Brasilia 70770-901, Brazil; ruben.romero2204@gmail.com

2   Laboratory of Biochemical Processes, EMBRAPA Agroenergia, Brasilia 70770-901, Brazil; luaserralas@gmail.com (L.A.S.); daianawischral@gmail.com (D.W.); joice.raisa@gmail.com (J.R.B.C.); thais.demarchi@embrapa.br (T.D.M.); thalyta.pacheco@embrapa.br (T.F.P.)

3   Laboratory of Genetics and Biotechnology, EMBRAPA Agroenergia, Brasilia 70770-901, Brazil

\*   Correspondence: felix.siqueira@embrapa.br (F.G.d.S.); joao.almeida@embrapa.br (J.R.M.d.A.)

**Abstract:** This work investigates the effects of the co-culture between the filamentous fungus *Panus lecomtei* and the yeast *Sporidiobolus pararoseus* in the production of laccases. The variations of time interval and inoculum volume of *S. pararoseus* in co-cultures with *P. lecomtei* stimulated laccase production, reaching its highest activity at nearly 2960.7 ± 244 U/mL with a maximum time point of 120 h and 2.0% (*v*/*v*), respectively. Further application in the pretreated sugarcane bagasse hydrolysis was performed, using *P. lecomtei* and *S. pararoseus* extract added to an enzyme mixture from the co-culture of *P. lecomtei* and *Trichoderma reesei* that positively favored the hydrolysis efficiency by 66.87%. Furthermore, the addition of *P. lecomtei* and *S. pararoseus* extract increased the degradation of industrial anthraquinone Remazol Brilliant Blue R by 78.98%. As a result, the extract derived from the co-culture of P. *lecomtei* and *S. pararoseus* rich in laccases presents potential in biotechnological applications, being suitable in the hydrolysis of lignocellulosic biomass and the degradation of unwanted dyes released in the environment.

**Keywords:** fungal co-culture; laccases; enzyme cocktail; lignocellulose; Remazol Brilliant Blue R

## 1. Introduction

Filamentous fungi secrete enzymes to obtain nutrients from plant sources during the decomposition/degradation process, playing a crucial role in maintaining the global carbon cycle. The fungi community can be subdivided according to the enzymatic profile utilized to degrade the plant cell wall. White-rot fungi are specialists in the high production of oxidative enzymes, such as laccases and peroxidases, responsible for modifying lignin and aromatic compounds. On the other hand, soft rot fungi are distinguished in the secretion of hydrolytic enzymes for example cellulases, hemicellulases, and pectinases for the degradation of cellulose, hemicellulose, and pectin, respectively. However, brown rot fungi can produce both hydrolytic and oxidative enzymes [1]. This classification has no taxonomic value, but it serves as a reference in the choice of one or more fungal species for a particular application at the industrial level. For instance, in the treatment or detoxification of xenobiotic compounds in the environment like synthetic dyes [2], and for the production of biofuels or models of plant biorefinery [3].

Enzyme production is one of the main applications of interspecific interactions of fungi in synthetic culture media. In competitive relationships, when two or more fungi

interact in direct contact, numerous changes are noticed at the morphological level or in the secretion of metabolites and enzymes [4,5]. Several studies have shown that co-cultures of two fungi with different enzymatic profiles, such as *Trichoderma spp* and white-rot fungi, can act in synergy to produce improved versions of these extracts, rich in cellulases and laccases [6]. In some cases, the overexpression of oxidative enzymes is detected in the co-cultures of filamentous fungi due to the removal of molecules that cause oxidative stress, for example, reactive oxygen species (ROS), generated as a defensive response via the cultivation partner [7].

The increase in enzymatic production of white-rot fungi has also been observed in co-cultures with more phylogenetically distant organisms, such as yeasts. *S. pararoseus* is considered a non-conventional yeast, with diverse applications in the industry, such as pharmaceuticals, cosmetics, and animal feed additives. This microorganism is capable of producing several metabolites of industrial interest, such as carotenoids, enzymes, exopolysaccharides, and others [8]. Previous studies demonstrated that the interaction between the white-rot fungi and *Trametes hirsuta* co-cultured with the yeast *S. pararoseus*, a well-known producer of β-carotenoids, in carbohydrate-rich culture media, led to an increase in laccase activity of up to 9.9 times higher than that of monocultures [9]. Furthermore, the co-cultures of *Pleurotus eryngii* var. *ferulae* with the yeasts *Rhodotorula mucilaginosa*, *Phaffia rhodozyma*, *S. pararoseus*, and *R. glutinis* resulted in higher laccase activities in the cell extracts [10,11]. Although studies of enzyme production improvement among filamentous fungi co-cultures are relatively numerous, co-cultures with yeast strains are rare.

*Panus lecomtei* is an edible white-rot fungi, isolated in the forests of the Amazon (Brazil). Preliminary studies showed that this fungus can grow in different types of residual plant biomass, like sugarcane bagasse, wheat bran, cottonseed cake, *Jatropha* cakes, empty fruit bunches, pressed fiber, and oil palm sludge. In addition, it can degrade toxic compounds such as gossypol and induce the high secretion of oxidative enzymes like laccases and peroxidases in solid and submerged fermentations [12,13]. In this work, the fungus *P. lecomtei* was co-cultured with the yeast *S. pararoseus* in submerged media supplemented with Oil Palm Decanter Cake (OPDC), wheat bran, and cottonseed cake as carbon sources aiming to increase laccase activity. The application of PlSp in the pretreated sugarcane biomass hydrolysis was evaluated in conjunction with an enzyme mixture from *T. reesei* and *P. lecomtei* (PlTr). Furthermore, the extract of PlSp was added in the reaction of decolorization of the anthraquinone dye Remazol Brilliant Blue R.

## 2. Materials and Methods

### 2.1. Fungi Strains and Biomass

The microorganism *Panus lecomtei* BRM 044603 belongs to the Collection of Microorganisms and Microalgae Applied to the Brazilian Agricultural Research Corporation (Embrapa Agroenergia, Brasilia, DF, Brazil). The filamentous fungus *Trichoderma reesei* RUT-C30 BRM 061104 and the yeast *S. pararoseus* ATCC16405/CCT 0779 were acquired from the American Type Culture Collection (ATTC) (Manassas, VA, USA).

For the biomasses, the company Denpasa SA (Pará, Brazil) kindly supplied the oil palm decanter cake (OPDC), which was constituted of cellulose 16.8%, hemicellulose 5.9%, lignin 17.4%, crude protein 15.3%, etheric extract 10.5% and ash 9.2%. The wheat bran and cottonseed were acquired from agricultural stores (Brasilia, DF, Brazil). For the preparation of the biomasses, these were dried at 65 °C for two days and milled with a micro-Wyllie type mill with a diameter of ≤2 mm. The Centro de Tecnologia Canavieira—CTC (São Paulo, SP, Brazil) provided the sugarcane bagasse. The pretreatment was performed with autohydrolysis with the following conditions: temperature of 192 °C, pressure of 10 bar, and duration of 38 min in a high-pressure reactor. The pretreated sugarcane bagasse will be abbreviated as PSB from now on.

*2.2. Production of Enzymatic Extracts of Mono or Co-Cultures of P. lecomtei, T. reesei and S. pararoseus*

To analyze the compatibility in vitro of fungal strains, *S. pararoseus* was placed together with *T. reesei* and *P. lecomtei* mycelia in potato dextrose (PDA) agar plates, separated approximately 4 cm for 9 days at 28 °C. Photographic records were made during the incubation days.

2.2.1. Mono or Co-Cultures of *Panus lecomtei* and *Trichoderma reesei*

For the cultivation, a liquid media based on the work of Peláez and collaborators [5] containing urea (0.45 g/L), yeast extract (0.75 g/L), bacteriological peptone (0.13 g/L), $(NH_4)_2SO_4$ (2.1 g/L), $KH_2PO_4$ (2.0 g/L), $CaCl_2$ (0.2 g/L), $MgSO_4.7H_2O$ (0.45 g/L), $CuSO_4.5H_2O$ (0.43 g/L), $ZnSO_4.7H_2O$ (1.4 mg/L), $MnSO_4.H_2O$ (1.04 mg/L) was prepared. The culture media was supplemented with the biomass OPDC (25 g/L), CS (50 g/L), and WB (20 g/L) as carbon sources. For monocultures, 5 mycelial discs (10 mm diameter each) of entirely colonized PDA agar (within 7 days of incubation) of *P. lecomtei* or *T. reesei* were employed as inoculum. In co-culture, 5 mycelial discs of entirely colonized agar of *P. lecomtei* were inoculated. After 24 h, each flask was inoculated with two mycelial discs of *T. reesei*. The cultivations were performed with three biological replicates in 125 mL flasks containing 50 mL of the culture media incubated at 25 °C, with an agitation of 150 rpm for thirteen days. Following this time interval, the cultures were transferred and centrifuged in 50 mL tubes at 10,600× *g* with a temperature of 5 °C for 10 min. The supernatant obtained was filtered with synthetic fabric to remove larger suspended particles from the liquid, which was called crude enzymatic extract. The co-culture was named PlTr.

2.2.2. Co-Culture of *Panus lecomtei* and *Sporidiobolus pararoseus*

The Yeast Malt Extract medium (YM) (glucose 10.0 g/L, peptone 5.0 g/L, malt extract 3.0 g/L, and yeast extract 3.0 g/L) was utilized for the cultivation of *S. pararoseus*. The cryopreserved cells were transferred to PDA agar plates and incubated at 28 °C for four days. Then, isolated colonies were inoculated in a 250 mL flask containing 30 mL of YM, incubated at 25 °C for 24 h and agitation of 120 rpm. After incubation, the culture was centrifuged at 2000× *g* for 5 min. The pellet was washed with sterile deionized water three times. After that, it was resuspended in water until an optical density (OD) close to 1.0 ($1 \times 10^6$ UFC/mL) was measured at 600 nm. In co-culture, 5 mycelial discs of fully colonized agar of *P. lecomtei* were inoculated in the media based on the work of Peláez et al. [5] and as described in Section 2.2.1. After 120 h, each flask was inoculated with 1 mL (2% *v/v*) of *S. pararoseus* inoculum. Cultivations were carried out in 125 mL flasks containing 50 mL of the culture media incubated at 25 °C, with an agitation of 150 rpm for thirteen days, and three biological replicates. The co-culture was named PlSp.

2.2.3. Co-Culture of *Panus lecomtei*, *Trichoderma reesei* and *Sporidiobolus pararoseus*

Submerged fermentation and co-culture inoculation of *P. lecomtei* and *T. reesei* were carried out in the same conditions described in Section 2.2.1. After seven days of incubation, 1 mL (2% *v/v*) of *S. pararoseus* inoculum was added to the flasks and cultivation continued at 25 °C, 150 rpm until thirteen days. The co-culture was named PlTrSp.

*2.3. Selection of S. pararoseus Inoculum Time Interval and Volume in Co-Culture with P. lecomtei on Laccase Production*

The submerged fermentation was carried out as described in Section 2.2.2. The evaluation of inoculum time interval and volume was carried out to evaluate the best conditions for laccase production. For the inoculum time interval, five mycelial discs of *P. lecomtei* were inoculated and immediately after, 1 mL (2% *v/v*) of *S. pararoseus* inoculum was added to the flasks (considered time 0 h). In different flasks, the same inoculum volume of *S. pararoseus* was added at 72, 120, and 168 h. The incubation parameters were a temperature of 25 °C for 13 days with an agitation of 150 rpm.

For the analysis of inoculum volume, five mycelial discs of *P. lecomtei* were inoculated and after 120 h, the volumes of 0.25, 0.5, 1.0, 2.0, 3.0, 4.0, and 5.0 mL were added in the 50 mL media, which represents 0.5, 1.0, 2.0, 4.0, 6.0, 8.0 and 10.0% (*v/v*) of *S. pararoseus* inoculum. The incubation was carried out at 25 °C for 13 days and agitation of 150 rpm. For both analyses (inoculum time and volume), a co-culture of *P. lecomei + T. reesei* (PlTr) and, monocultures of *P. lecomtei* (Pl), *T. reesei* (Tr), and *S. pararoseus* (Sp) were performed simultaneously.

### 2.4. Enzymatic Assays

The total cellulases (Fpase) and laccase activities were performed using the enzymatic extracts obtained from mono or co-cultures detailed in Section 2.2. The Fpase assay was carried out with Whatman® No. 1 filter paper (GE Healthcare, Amershan, UK) as a substrate, and the reducing sugars were determined using the dinitrosalicylic acid (DNS) method [14,15]. One unit of Fpase was defined as the amount of enzyme that hydrolyzes the substrate to generate 1 μmol of glucose per minute.

The activity of laccase was detected via the oxidation of 2,2′-azino-bis (3-ethylbenzothiazoline-6-sulfonic acid) (ABTS) 5 mM at 420 nm [16,17]. One laccase unit was defined as the amount of enzyme that oxidizes ABTS to generate 1 μmol of oxidized ABTS per minute.

### 2.5. Hydrolysis of Pretreated Sugarcane Bagasse by the Co-Cultures Enzymatic Cocktails

The capability to decompose lignocellulosic biomass using the enzymatic extracts produced via mono- or co-culture was evaluated using pretreated sugarcane bagasse (PSB). The hydrolysis of this substrate was carried out in 50 mL flasks holding PSB adjusted at 5% *w/v* (dry mass) and 7 mL of citric acid/sodium citrate buffer 0.5 M pH 5.0. Four mixed enzyme cocktails were verified: 1. 10.5 mL PlTr extract + 4.5 mL deionized water; 2. 10.5 mL PlTr extract + 4.5 mL PlSp (5% *v/v* inoculum size); 3. 10.5 mL PlSp (5% *v/v* inoculum size) + 4.5 mL deionized water; 4. 10.5 mL Pl (monoculture) + 4.5 mL deionized water; 5. 15 mL of deionized water (enzyme-free control). The mixed cocktails were compared with assays using 10.5 mL of extracts from Tr, Pl, PlTr, PlTrSp and PlSp individually. The flasks were incubated at 50 °C with an agitation of 200 rpm for 48 h.

Hereafter, 0.5 mL aliquots were collected and centrifuged at 14,000× *g* for 5 min. The sugar concentration (glucose and xylose) from the supernatants was quantified via HPLC equipped with an HPX-87H column. For the mobile phase, 5 mM $H_2SO_4$ was applied, with a column temperature of 45 °C, and a flow rate of 0.6 mL/min. The calculations of hydrolysis efficiency (%) were based on the conversion rate of cellulose and hemicellulose into glucose and xylose [18]. All the equations can be found in detail in the work of Peláez et al. (2022) [5]. Compositional analysis of PSB was determined following the protocol recommended by the National Renewable Energy Laboratory [19]. The composition of PBS was determined as cellulose 16.8%, hemicellulose 5.9%, lignin 17.4%, crude protein 15.3%, etheric extract 10.5% and ash 9.2%.

### 2.6. Decolorization of Dye Remazol Brilliant Blue R with Enzyme Extracts

Remazol Brilliant Blue R stock solution was prepared using a final concentration of 100 mg/mL in a sodium acetate buffer at 100 mM pH 5.0. The test was adjusted from the methodology described in Yadav et al. [20]. In 96-well microplates, 300 μL of Remazol Brilliant Blue R together with 100 μL of Sp (5% *v/v* inoculum size) were added. The same volumes to evaluate the PlSp, PlTr, *T. reesei + S. pararoseus* (TrSp), co-cultures, *P. lecomtei* and *T. reesei* monoculture, and the control (only water) were utilized. The reaction was carried out at 35 °C for 4 h, in a UV-VIS spectrophotometer at 592 nm. The percent dye fade was calculated using the following formula:

$$Decolorization\ of\ RBBR = \frac{(A_{Initial} - A_{Final})}{A_{Initial}} \times 100 \tag{1}$$

where, $A_{Initial}$ and $A_{Final}$ are the absorbances of Remazol Brilliant Blue R + Enzyme at 0 h and 4 h of reaction, respectively.

### 2.7. Statistical Analysis

The calculations were carried out in Excel®, whereas statistical analysis such as the Tukey test (q < 0.05), and ANOVA were developed at Statistica® 7.0, and SISVAR® software version 5.6, respectively. All experiments were performed in biological and technical triplicate.

## 3. Results

### 3.1. Growth Interaction of S. pararoseus with T. reesei and P. lecomtei

The growth interaction patterns of *S. pararoseus*, *T. reesei*, and *P. lecomtei* were characterized by the formation of pigmentation zones related to an oxidative stress response mediated via the release of oxidative enzymes for instance laccases (Figure 1) [21]. The growth pattern between *S. pararoseus* and *P. lecomtei* did not display visible barrier formation or pigmentation. Instead, *P. lecomtei* exhibited invasive growth on the surface of *S. pararoseus* (Figure 1e). On the other hand, the interaction between *T. reesei* and *S. pararoseus* was marked by the formation of a presumed inhibition line. Nevertheless, at the end of the incubation time *T. reesei* grew invasively on *S. pararoseus*, as well as *P. lecomtei* (Figure 1f).

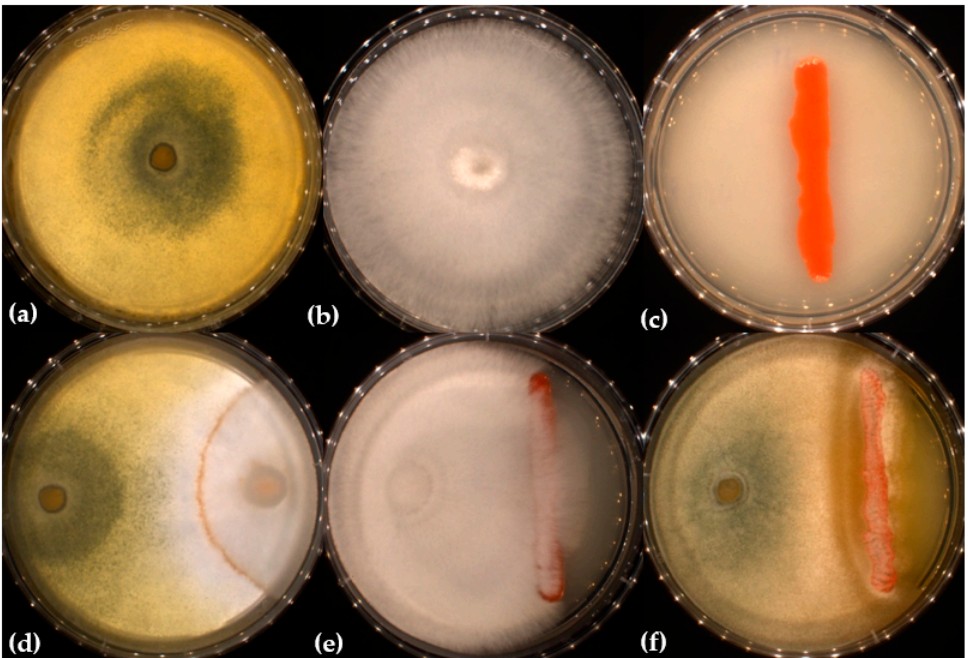

**Figure 1.** Growth interaction between *S. pararoseus*, *T. reesei* and *P. lecomtei* on potato dextrose agar (PDA) plates. (**a**) growth pattern of *T. reesei* BRM 061104 (Tr), (**b**) *P. lecomtei* BRM 044603 (Pl), (**c**) *S. pararoseus* (Sp) and their growth interactions, (**d**) *P. lecomtei* + *T. reesei* (PlTr), (**e**) *P. lecomtei* + *S. pararoseus* (PlSp) and *T. reesei* + *S. pararoseus* (**f**).

### 3.2. Effect of Inoculum Time Interval and Volume of S. pararoseus in Co-Culture with P. lecomtei on Laccase Production

The effect of the variation in the co-culture conditions of *S. pararoseus* and *P. lecomtei* impacted the performance in laccase production. The modifications of the time interval between the inoculum and its volume (% *v/v*) generated the results shown in Figure 2. Laccase activity was higher in the enzyme extract with a 120h time interval inoculum, yielding 2960.7 ± 244 U/mL (Figure 2a). The treatment of 72 h also indicated a positive effect of enhancing laccase activity, reaching 2532.8 ± 282 U/mL. Thus, the co-culture of *P. lecomtei* and *S. pararoseus* required an inoculum time interval ranging from 72 to 120 h to enhance laccases. In contrast, cellulase activities of PlSp co-cultures did not suggest stimulation regarding time interval (Figure 2b). Based on statistical analysis, cellulases

were lower in co-culture compared to the Pl monoculture, which indicates that this strategy is not recommended for cellulase increase in fungal species like *P. lecomtei*.

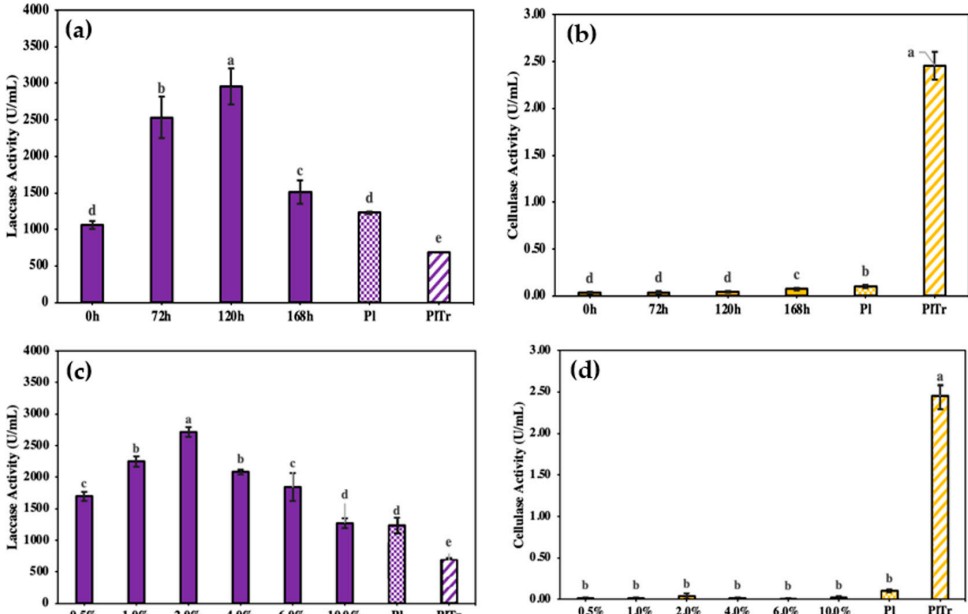

**Figure 2.** Effect of inoculum interval time and volume size of *S. pararoseus* on enzyme activity of extracts obtained from the co-culture of *P. lecomtei* and *S. pararoseus*. *Co*-cultures of *P. lecomtei* and *S. pararoseus* are shown in fully colored bars, whereas *P. lecomtei* + *T. reesei* (PlTr) are shown in lined bars, and monoculture of *P. lecomtei* (Pl) in checkered bars. (**a**) Laccase activity with different inoculum times, (**b**) Cellulase activity with different inoculum times, (**c**) Laccase activity with seven inoculum sizes, and (**d**) Cellulase activity of seven inoculum sizes. Different letters mean significant differences (Tukey test *p* < 0.05).

Fixing the inoculum interval time in 120 h, the volume of *S. pararoseus* inoculum in the co-culture with *P. lecomtei* also influenced laccase activity (Figure 2c). The highest activity was observed with the *S. pararoseus* inoculum at 2.0% (*v*/*v*), producing 2713.8 ± 73 U/mL. Furthermore, inoculum volumes higher or lower than 2.0% (*v*/*v*) suggested less laccase activity in the co-culture. Regarding the cellulases, the inoculum volume of *S. pararoseus* did not exhibit a significant increase (Figure 2d). As a result, for maximum laccase production, the extract with an inoculum interval time of 120 h and volume of 2.0% (*v*/*v*) was determined as optimized.

### 3.3. Enzymatic Activities of Mono and Co-Cultures Extracts

The investigation of the effects of co-cultivating *S. pararoseus* with *T. reesei* and *P. lecomtei* on cellulases and laccase production was explored and summarized in Figure 3. Laccase activities (Figure 3a) displayed significant differences between PlSp and PlTr co-cultures with the Pl monoculture. The PlSp extract exhibited the highest laccase activity (2290.6 ± 49 U/mL), which is 2.1-fold higher than the monoculture of *P. lecomtei* (1079.5 ± 95 U/mL). However, the laccase activity of the PlTr extract (857.3 ± 109 U/mL) was lower than the Pl monoculture. This fact strongly indicates the positive relation between the production of laccase and the yeast-fungus interaction of *P. lecomtei* and *S. pararoseus*. Also, laccase enhancement is directly related to *P. lecomtei* being the main productor of this enzyme since *T. reesei* and *S. pararoseus* monocultures or their co-culture did not have activity. Nevertheless, this stimulatory laccase induction was not observed with the PlTrSp co-culture, with a value that was significantly lower than the PlSp co-culture but similar to the Pl monoculture (*p* < 0.05) (Figure 3a). In this case, the simultaneous culture between the three fungi did not represent an interesting strategy to enhance laccases.

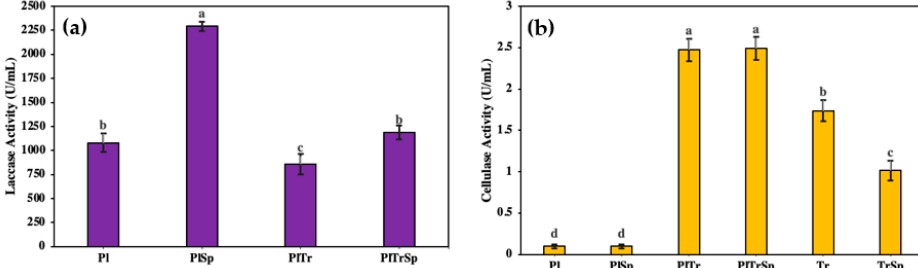

**Figure 3.** Enzymatic activities of laccase and cellulases of monocultures and co-cultures. (**a**) Laccase activity, (**b**) cellulase activity. Legends: *P. lecomtei* + *S. pararoseus* (PlSp), *P. lecomtei* + *T. reesei* (PlTr), *T. reesei* + *S. pararoseus* (TrSp), *P. lecomtei* + *T. reesei* + *S. pararoseus* (PlTrSp), and monocultures of *P. lecomtei* (Pl) and *T. reesei* (Tr). Different letters mean significant differences (Tukey test *p* < 0.05).

The induction of cellulases in co-cultures reached the maximum activity with both co-cultures PlTr and PlTrSp, yielding 2.47 ± 0.13 and 2.49 ± 0.13 U/mL, respectively (Figure 3b). The addition of *S. pararoseus* on the 7th day of PlTr incubation did not affect the cellulase secretion performance of PlTr. Moreover, the addition of *S. pararoseus* did not enhance cellulase secretion in the PlSp co-culture (<0.09 U/mL) and decreased this enzyme production of *T. reesei* by 1.7 times. Contrary to laccases, the cellulases from *P. lecomtei* did not respond positively to the cultivation conditions carried out with *S. pararoseus*. Also, the addition of the yeast in conjunction with *T. reesei* negatively affected the cellulase production (Figure 3b).

### 3.4. Hydrolytic Performance of Co-Cultures Enzymatic Cocktails

The enzymatic hydrolysis efficiency of PBS indifferent extracts (without mixing) demonstrated a similar behavior of cellulase activities, where the maximum hydrolysis efficiency of both cellulose and hemicellulose was observed in PlTr (66.4 ± 5% and 45.5 ± 2%, respectively) (Figure 4a). When the enzyme extract PlTrSp was applied, it resulted in a 10% lower efficiency when compared to PlTr. Therefore, the overall bioconversion of cellulose and hemicellulose was not increased by preparing an enzyme cocktail of three fungi in the same batch. However, since laccases play a central role in the bioconversion of lignocellulosic biomass, the enzyme extract with the highest laccase activity (PlSp) and the one with the highest hydrolysis cellulose efficiency (PlTr) were blended.

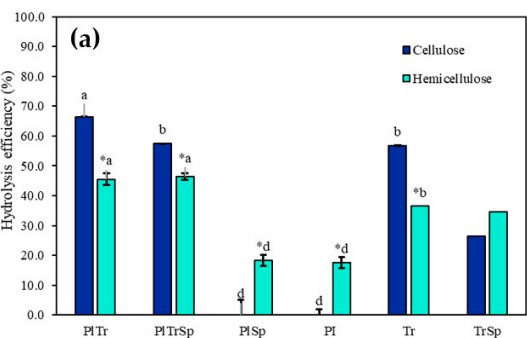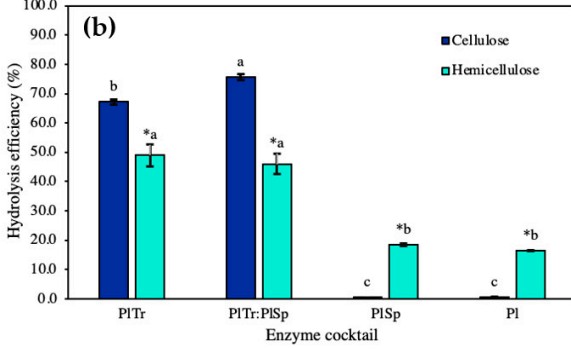

**Figure 4.** Hydrolysis efficiency of cellulose and hemicellulose in pretreated sugarcane bagasse (PBS) using different extracts alone or in combination. (**a**) Hydrolysis using the enzymatic extract from the same batch cultures without mixing it. Monocultures of *P. lecomtei* (Pl) and *T. reesei* (Tr). Co-cultures of *P. lecomtei* and *T. reesei* (PlTr), *P. lecomtei* + *T. reesei* and *S. pararoseus* (PlTrSp), and *T. reesei* and *S. pararoseus* (TrSp). (**b**) Hydrolysis using blended extracts enzymes in different proportions. The treatments were as follows: *P. lecomtei* + *T. reesei* with water (PlTr, ratio 70:30); *P. lecomtei* + *T. reesei* supplemented with *P. lecomtei* + *S. pararoseus* (PlTr:PlSp, ratio 70:30); *P. lecomtei* + *S. pararoseus* with water (PlSp, ratio 70:30); and *P. lecomtei* with water (Pl, ratio 70:30). Different letters mean significant differences (Tukey test *p* < 0.05). * Statistical data for hemicellulose.

The hydrolytic potential of the PlTr extract and the oxidative potential rich in laccase of the optimized PlSp were investigated as a potential enzyme cocktail capable of bioconverting PSB into sugar monomers efficiently. For this, the enzyme load of PlTr was reduced to 70%, and the remaining 30% was enriched with PlSp laccases. The results were compared with the PlTr extract without the laccases, the PlSp extract, and the Pl monoculture at 70%. The cellulose hydrolytic efficiency of the PlTr/PlSp cocktail was $75.69 \pm 1.1\%$ corresponding to 11.2% higher than the PlTr extract, which yielded $67.23 \pm 0.8\%$, as seen in Figure 4b. The preparations of PlSp and Pl alone did not exhibit a significant effect on cellulose hydrolysis.

For hemicellulose, there was no statistical difference between the hydrolysis efficiency of the PlTr extract and PlTr/PlSp cocktail, which was next to 50% for both treatments. The results for PlSp and Pl were similar, indicating approximately 20% of hemicellulose hydrolysis efficiency. The overall hydrolysis efficiency of the enzyme cocktail PlTr/PlSp, including the bioconversion of cellulose and hemicellulose, yielded 66.87%.

### 3.5. Effect of Enzyme Extracts in the Decolorization of the Dye Remazol Brilliant Blue R

The optimized PlSp extract was evaluated regarding the application of oxidative enzymes as potential agents for the degradation of the industrial dye Remazol Brilliant Blue R. In this sense, besides PlSp, the extract from PlTr, and Pl alone were also assessed as possible agents for the decolorization of this dye, as seen in Figure 5.

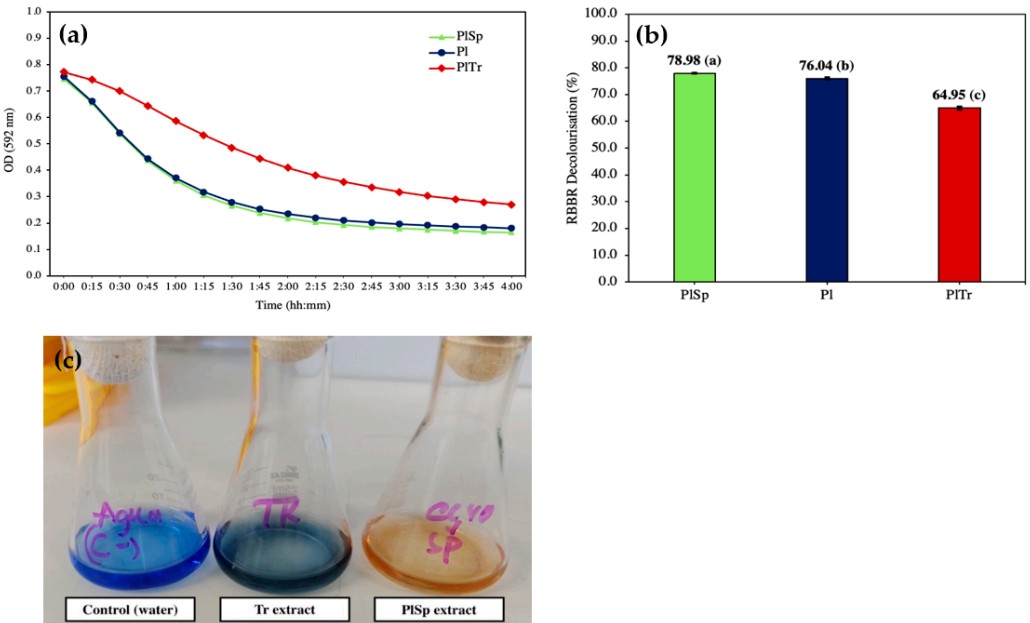

**Figure 5.** Effect of different extracts on the Remazol Brilliant Blue R degradation. (**a**) Remazol Brilliant Blue R decolorization kinetic of *P. lecomtei + S. pararoseus* (PlSp), *P. lecomtei + T. reesei* (PlTr), and *P. lecomtei* alone (Pl). (**b**) Remazol Brilliant Blue R decolorization (%) after 4 h of reaction by using enzymatic extracts of PlSp, PlTr, and Pl. (**c**) Color grades of Remazol Brilliant Blue R degradation by the control (water), treatment with an extract without laccase (Tr), and PlSp enzyme extract rich in laccases. Different letters mean significant differences (Tukey test $p < 0.05$).

The effect of PlSp and Pl alone was similar in the discoloration kinetic of Remazol Brilliant Blue R, starting with an OD next to 0.8 and finishing at 0.2. The PlTr extract had a lower effect on the discoloration compared to the other two extracts since its final OD was higher than 0.3 (Figure 4a). The maximum decolorization rate of Remazol Brilliant Blue R was $78.98 \pm 0.01\%$ utilizing the PlSp enzyme extract, which holds the highest laccase activity. The enzyme extract of Pl monoculture obtained similar results reaching a value close to $76.04 \pm 0.01\%$, while the PlTr yielded $64.95 \pm 0.01\%$ (Figure 5b).

When extracts from *T. reesei* + *S. pararoseus* (TrSp) and *T. reesei* (Tr) alone were analyzed, they did not display Remazol Brilliant Blue R decolorization response. This fact is possibly related to laccase activities playing the principal role in this reaction. For instance, looking at Figure 5c, it is possible to notice that the Tr extract did not exhibit discoloration compared to the control. Yet, adding PlSp which is rich in laccases, exhibited a significant difference in color compared to the control and Tr. Moreover, this is the first work reporting laccase usage with a considerable dye decolorization obtained from yeast fungi co-cultures.

## 4. Discussion

The co-cultures of fungi respond categorically to some variables, such as different temperatures, time intervals between inoculum, and medium composition for enzyme production, for example laccase and cellulase [5]. From the results obtained in this work, the time interval and volume of the inoculum of the yeast *S. pararoseus* directly influenced laccase production. Similarly, in the work of Zhang et al., *T. hirsuta* was co-cultivated with *S. pararoseus* inoculated after seven days of cultivation with a volume of 2% (*v/v*). This interaction resulted in higher laccase activity (31.777 U/L) after completing 12 days in co-culture, increasing by 9.9 times compared to the monoculture of *T. hirsuta* [9]. Contrary to their study, prolonging the time interval for more than 168 h negatively affected the laccase secretion of *P. lecomtei*, while simultaneous culture (0 h time interval) showed no enhancement.

Although few reports investigated the laccase improvement by cultivating filamentous fungi and yeast, some hypotheses of these beneficial effects are being raised. First, the glucose starvation generated via yeast growth and, second, the glycerol produced by this microorganism could have influenced positively laccase production. Also, the β-carotene produced by yeasts, such as *S. pararoseus*, may act as a stimulatory compound inducing laccase secretion [9–11,22]. For instance, other β-carotene-producing yeasts like *Rhodotorula mucilaginosa*, *R. glutinis*, and *Phaffia rhodozyma* demonstrated an induction for the laccase production of *Pleurotus eryngii* var. *ferulae* by 2.0 to 3.0 times [11].

β-Carotene is exploited as an antioxidant agent in the food industry. There are a few reports that considered the use of these agents as inducers for laccase production, for example, ferulic acid, guaiacol, and other phenolic compounds [11,23]. When they are added to the culture, oxygen radicals are generated, triggering a response to oxidative stress. Consequently, with elevated volumes of intracellular ROS, like hydrogen peroxide ($H_2O_2$) and hydroxyl radical ($OH^-$), laccase production is enhanced [24]. Contrary to the laccases, cellulases from *P. lecomtei* did not seem to respond to stimulatory compounds, such as β-carotene, in the cultivation conditions carried out with *S. pararoseus*, and it did affect negatively when cultivated with *T. reesei* alone. However, there are reports stating that both ROS and β-carotene could influence positively cellulase production, even though the mechanism is still not fully understood [25–27]. Therefore, further studies are necessary to elucidate the relationship between these compounds and cellulase production in the co-culture of TrSp and PlSp.

In this sense, the addition of highly active laccases played a fundamental role in the hydrolysis of sugarcane bagasse by removing lignin fragments that prevent cellulase access to cellulose [28]. Enzymatic cocktails supplemented with extracts rich in laccases produced by basidiomycetes, including commercial enzymes, have shown an increase in the hydrolysis efficiency of lignocellulosic biomass. In the study of Matei et al., an enzyme extract produced by *Trametes villosa* (laccase activity of 9467.8 U/L) supplemented the commercial extract Cellic Ctec 2® (Novozymes, Bagsværd, Denmark) for the hydrolysis of sugarcane bagasse. In the reaction, Ctec2 blended with the extract had an increase from 155.02 to 192.45 mg/g of dry biomass of reducing sugars [29]. The blend of an extract abundant in cellulases with another rich in laccases, like PlTr/PlSp cocktail, allowed the increase in cellulose bioconversion efficiency when compared to previous results [5,30], even with a 30% reduction in the maximum enzyme load of extract PlTr. Moreover, considering that PlTr/PlSp cocktail is a crude extract without the addition of enzyme stabilators or chemical



additives, like commercial enzymes, PlTr/PlSp demonstrates great potential to be applied in several hydrolytic processes of lignocellulosic biomass conversion. Also, further studies could be conducted aiming to scale up PlTr/PlSp from an industrial perspective.

Another application for the PlSp extract was considered since it is estimated that synthetic dyes destined for the textile industry are produced at a scale of $7 \times 10^7$ tons globally. At least 10% of these compounds are discharged into clean water, causing a severe environmental problem [31,32]. The anthraquinone dyes, which include the Remazol Brilliant Blue R, are among the most important classes of colorants used in textile industries and are considered a model for the evaluation of dyes bioremediation. The use of peroxidases for this purpose is being investigated since these enzymes have considerable plasticity in different substrates [33]. The extract obtained from the co-culture of PlSp exhibited promising results reaching almost 80% in the decolorization rate of Remazol Brilliant Blue R. Different studies demonstrated that fungal enzymes are capable of degrading this dye at a decolorization rate ranging from 50 to 80% [34–37]. Therefore, PlSp displayed great potential in this application with values comparable to the literature.

Additionally, one of the main findings in the application of fungi extracts is their rapid decolorization rate on Remazol Brilliant Blue R, reaching a high percentage value within 4 h. In this study, the decolorization test was monitored for 48 h but OD remained the same, which means that a significant reaction occurred in the first 4 h. Higher values of decolorization of Remazol Brilliant Blue R depend on different variables, including the potential redox and the specific characteristic on the active site of laccases, or other oxidative enzymes non-quantified in this study, such as the dye-decolorizing peroxidase, and the structural complexity of the dyes [38–40]. Therefore, further studies can be explored on these variables to improve the bioremediation of textile dyes.

## 5. Conclusions

Fungal interactions influence the production of lignocellulolytic enzymes, which can be applied in processes of industrial importance, such as the bioconversion of structural components of plant biomass and the bioremediation of dyes. The interactions between the basidiomycete *Panus lecomtei* and the yeast *Sporidiobolus pararoseus* positively impact laccase production, reaching an activity of up to 2900 U/mL. The time interval and volume of *S. pararoseus* inoculum influence the improvement of laccase secretion. Finally, the elaboration of an enzymatic cocktail from co-culture extracts of PlTr and PlSp allowed cellulose conversion of up to 75%. Moreover, the PlSp was the first co-culture extract to reach almost 80% of the Remazol Brilliant Blue R decolorization rate, demonstrating the potential to be applied as a bioremediation agent in the textile industry.

**Author Contributions:** Conceptualization, R.D.R.P., F.G.d.S. and J.R.M.d.A.; methodology, R.D.R.P., T.D.M. and T.F.P.; formal analysis, R.D.R.P., T.D.M. and T.F.P.; investigation, R.D.R.P., D.W., J.R.B.C. and F.G.d.S.; data curation, R.D.R.P., L.A.S., D.W., T.D.M. and T.F.P.; writing—original draft preparation, R.D.R.P.; writing—review and editing, L.A.S., F.G.d.S. and J.R.M.d.A.; visualization, F.G.d.S. and J.R.M.d.A.; supervision, F.G.d.S. and J.R.M.d.A.; project administration, F.G.d.S.; funding acquisition, F.G.d.S. and J.R.M.d.A. All authors have read and agreed to the published version of the manuscript.

**Funding:** This research was funded by the Brazilian Agricultural Research Corporation (EMBRAPA), Federal District Research Support Foundation (FAP-DF, 0193.001720-2017), and supported by the Coordination for the Improvement of Higher Education Personnel (CAPES), Brasília, Brazil.

**Institutional Review Board Statement:** Not applicable.

**Informed Consent Statement:** Not applicable.

**Data Availability Statement:** Not applicable.

**Acknowledgments:** The authors would like to thank CAPES (Coordination for the Improvement of Higher Education Personnel), the University of Brasília (UnB), the Federal District Research Support Foundation (FAP-DF), and the Brazilian Agricultural Research Corporation (EMBRAPA), Brasília, Brazil, for their financial support.

**Conflicts of Interest:** All authors were employed by the company Embrapa. All authors declare that the research was conducted in the absence of any commercial or financial relationships that could be construed as potential conflicts of interest. The authors declare that this study received funding from Embrapa, FAP, and Capes. The funders had the following involvement with this study: financial support.

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
