# Peer review of "Improvement of Laccase Activity in Co-Culture of Panus lecomtei and Sporidiobolus pararoseus and Its Application as an Enzymatic Additive in Biomass Hydrolysis and Dye Decolorization"

_fermentation, doi:10.3390/fermentation9110945_

Round 1

Reviewer 1 Report

Comments and Suggestions for Authors

The article is devoted to an interesting issue - the study of enzyme production by co-cultivated fungal cultures. Such studies are interesting both from a practical point of view and from the point of view of studying microbial communities.

p. 3 line 108 How long does it take for completely colonized agar by fungi? This should have been indicated.

Enzyme extracts were obtained by filtration through a paper filter (p. 3, line 115). Could sorption of cellulases occur on paper, since cellulases have an affinity for cellulose?

What is the mechanism for the reduction in cellulose production in the co-culture of S. pararoseus and T. reesei? It is unlikely that this can be explained only by the possible action of β-carotene. This mechanism should be discussed.

Since the decomposition of hemicellulose is being studied, it would be interesting to have data on the production of hemicellulases (xylanases, beta-glucanases, etc.) by fungi.

Author Response

The article is devoted to an interesting issue - the study of enzyme production by co-cultivated fungal cultures. Such studies are interesting both from a practical point of view and from the point of view of studying microbial communities.

Response: We would like to express our thanks for the comments which contributed to the improvement of the manuscript. All points raised by your revision were addressed in the following and the modifications were incorporated into the revised manuscript.

Comment 1: p. 3 line 108 How long does it take for completely colonized agar by fungi? This should have been indicated.

Response: It took 7 days for the complete colonization of the agar by the fungi (Panus lecomtei and Trichoderma reesei). This information was added to the text, as pointed out by the reviewer (p.3, line 122).

Comment 2: Enzyme extracts were obtained by filtration through a paper filter (p. 3, line 115). Could adsorption of cellulases occur on paper, since cellulases have an affinity for cellulose?

Response: We agreed with the comment and the revised text was added to the manuscript (p.3, line 130). Indeed, the adsorption of cellulases could occur using a paper filter. We made a mistake in stating that filter paper was used in filtration, when in fact, it was only used in enzymatic hydrolysis (FPase activity). The supernatant was filtered through a synthetic fabric to separate larger particles from the liquid and generate the crude enzymatic extract. We thank the reviewer for pointing this out.

Comment 3: What is the mechanism for the reduction in cellulose production in the co-culture of S. pararoseus and T. reesei? It is unlikely that this can be explained only by the possible action of β-carotene. This mechanism should be discussed.

Response: We thank the reviewer for the comment and suggestion. In fact, the β-carotene could act in a positive response for cellulase production, as well as reactive oxygen species (ROS). However, studies have shown that the mechanism is not fully elucidated yet. That is why further, and extensive studies should be done regarding the relationship of these compounds and cellulase production in the co-cultivation of T. reesei + S. pararoseus, and P. lecomtei + S. pararoseus. This topic was added in the discussion section (p. 10, lines 404 – 408).

Comment 4: Since the decomposition of hemicellulose is being studied, it would be interesting to have data on the production of hemicellulases (xylanases, beta-glucanases, etc.) by fungi.

Response: We thank the reviewer for the comment and suggestion. The focus of this work was to evaluate and increase the laccase activity, through optimization of the co-cultivation of two fungi. Furthermore, the potential application of these laccase-rich extracts in biomass saccharification and dye decolorization was investigated. Therefore, to study in detail the production of hemicellulases, the effect of co-cultures on their production, and application, further studies focused on that must be carried out.

Reviewer 2 Report

Comments and Suggestions for Authors

General.

The manuscript describes production of laccases by co-cultivation of a filamentous fungus and a yeast strain. The manuscript is well written with minor spelling errors. The authors cite 38 articles, which seem relevant for the introduction to the study.  A few self-citations are also relevant.

Some of the abbreviations are not standard abbreviations, please use whole names of f. inst white-rot fungi, do not use WRF. Remazol brilliant blue R should not be abbreviated to RBBR, use whole term, also in keywords. Reactive organic species is abbreviated ROS, which is correct, this is a standard, well known abbreviation.

Abstract.

The text from line 13 to 18 belongs to the Introduction section and should be moved there. The abstract should present the most important results and also should not contain experimental data as presented in parentheses. General abbreviations should not appear  in an abstract, but in the main text.

Citations should be presented as Peláez et al. (et al. in italics)

Author Response

The manuscript describes the production of laccases by co-cultivation of a filamentous fungus and a yeast strain. The manuscript is well written with minor spelling errors. The authors cite 38 articles, which seem relevant for the introduction to the study. A few self-citations are also relevant.

Response: We would like to express our thanks for the questions and comments which contributed to the improvement of the manuscript. All points raised by your revision were addressed in the followings and the modifications were incorporated into the revised manuscript.

Comment 1: Some of the abbreviations are not standard abbreviations, please use whole names of f. inst white-rot fungi, do not use WRF. Remazol brilliant blue R should not be abbreviated to RBBR, use whole term, also in keywords. Reactive organic species is abbreviated ROS, which is correct, this is a standard, well known abbreviation.

Response: We agreed with the comment, and the abbreviations throughout the abstract (p. 1, line 19 – 26) keywords (p. 1, line 30), and text were erased (p.1, lines 37,39,42; p.2, line 52, 58, 64, 72; p. 5, line 206 – 210, 218; p. 9, line 346 – 360, 369 – 372; p. 10, line 428; p.11, line 433, 435, 438, 441, 456).   

Comment 2: Abstract. The text from line 13 to 18 belongs to the Introduction section and should be moved there. The abstract should present the most important results and also should not contain experimental data as presented in parentheses. General abbreviations should not appear in an abstract, but in the main text.

Response: We agreed with the comment and the revised abstract was updated (p. 1, lines 16 – 28).

Comment 3: Citations should be presented as Peláez et al. (et al. in italics).

Response: The citations containing et al. throughout the text were corrected (p. 3, line 143; p. 5, line 209; p. 10, line 382; p. 10, line 413).